# Transcutaneous Spinal Cord Stimulation Improves Respiratory Muscle Strength and Function in Subjects with Cervical Spinal Cord Injury: Original Research

**DOI:** 10.3390/biomedicines11082121

**Published:** 2023-07-27

**Authors:** Hatice Kumru, Loreto García-Alén, Aina Ros-Alsina, Sergiu Albu, Margarita Valles, Joan Vidal

**Affiliations:** 1Institut Guttmann, Institut Universitari de Neurorehabilitació Adscrit a la (UAB), 08916 Barcelona, Spain; loretogarcia@guttmann.com (L.G.-A.); superaina@gmail.com (A.R.-A.); salbu@guttmann.com (S.A.); mvalles@guttmann.com (M.V.); jvidal@guttmann.com (J.V.); 2Universitat Autònoma de Barcelona, Cerdanyola del Vallès, Bellaterra, 08193 Barcelona, Spain; 3Fundació Institut d’Investigació en Ciències de la Salut Germans Trias i Pujol, 08916 Barcelona, Spain

**Keywords:** transcutaneous electrical spinal cord stimulation, cervical spinal cord injury, respiratory function, rehabilitation

## Abstract

(1) Background: Respiratory muscle weakness is common following cervical spinal cord injury (cSCI). Transcutaneous spinal cord stimulation (tSCS) promotes the motor recovery of the upper and lower limbs. tSCS improved breathing and coughing abilities in one subject with tetraplegia. Objective: We therefore hypothesized that tSCS applied at the cervical and thoracic levels could improve respiratory function in cSCI subjects; (2) Methods: This study was a randomized controlled trial. Eleven cSCI subjects received inspiratory muscle training (IMT) alone. Eleven cSCI subjects received tSCS combined with IMT (six of these subjects underwent IMT alone first and then they were given the opportunity to receive tSCS + IMT). The subjects evaluated their sensation of breathlessness/dyspnea and hypophonia compared to pre-SCI using a numerical rating scale. The thoracic muscle strength was assessed by maximum inspiratory (MIP), expiratory pressure (MEP), and spirometric measures. All assessments were conducted at baseline and after the last session. tSCS was applied at C3-4 and Th9-10 at a frequency of 30 Hz for 30 min on 5 consecutive days; (3) Results: Following tSCS + IMT, the subjects reported a significant improvement in breathlessness/dyspnea and hypophonia (*p* < 0.05). There was also a significant improvement in MIP, MEP, and forced vital capacity (*p* < 0.05). Following IMT alone, there were no significant changes in any measurement; (4) Conclusions: Current evidence supports the potential of tSCS as an adjunctive therapy to accelerate and enhance the rehabilitation process for respiratory impairments following SCI. However, further research is needed to validate these results and establish the long-term benefits of tSCS in this population.

## 1. Introduction

Impaired respiratory function is a common consequence of cervical spinal cord injury (cSCI) and may also occur following thoracic injuries [1,2,3]. A total of 84% of cervical (C1-4), 60% of C5-8, and 65% of thoracic (Th1-12) injuries had respiratory complications [1]. It affects ventilation and lung volumes, leading to breathlessness or dyspnea and an increased work of breathing [2,3]. A disruption of supraspinal input to intercostal and abdominal motor neurons can lead to reduced respiratory capacities, peak expiratory flow, and coughing strength. An impairment of the muscles of inspiration reduces vital capacity, prevents deep breaths, and may lead to dyspnea with exertion and/or collapse of the lungs (atelectasis). Dysfunctional expiratory muscles impair cough and secretion clearance, increase airways resistance and increase the susceptibility to and persistence of lower respiratory tract infections. Total lung capacity is usually reduced following SCI due to impaired inspiratory musculature and residual volume is relatively increased due to impaired expiratory musculature and subsequent reduced expiratory reserve volume [2,3,4]. Impaired respiratory function is a significant contributor to the overall morbidity and mortality of SCI individuals [5,6]. Thus, any intervention that could be shown to improve respiratory function and thereby prevent or alleviate these conditions would be of great benefit to the SCI population. On a day-to-day basis, if respiratory muscle training is effective for a person with SCI, it could mean breathing and speaking more easily, coughing and expelling mucus more effectively, and avoiding hospital admissions [2,4]. Each of these outcomes (both individually and in combination) could significantly improve such a person’s quality of life.

Respiratory muscle training (RMT) is a non-invasive intervention that can improve respiratory muscle function and reduce the risk of respiratory complications in individuals with SCI, and is widely used in our hospital. RMT involves exercises that target the inspiratory and expiratory muscles to improve strength (maximal inspiratory and expiratory pressure: MIP and MEP), endurance, and coordination [2,3]. In addition to RMT, other non-invasive interventions in individuals with SCI include breathing exercises, airway clearance techniques, and positioning. These interventions aim at optimizing lung volume, reducing airway resistance, and promoting efficient breathing patterns [2,3]. The other current invasive neuromodulation techniques, including phrenic nerve pacemaker (PNP) [7] and a technique known as diaphragm motor point pacing (DMPP), have been developed and successfully tested [8] specifically in individuals with lesions above the C4 level [9]. DMPP is a technique that involves the implantation of electrodes near the motor points of the diaphragm to stimulate the diaphragm muscles, thereby improving respiratory function in people with spinal cord injury, used at the Guttmann Institute. In addition to these invasive techniques, epidural spinal cord stimulation has also shown promising results in improving respiratory function following SCI [10].

Transcutaneous electrical spinal cord stimulation (tSCS) is a non-invasive spinal cord stimulation technique promoting the functional and motor strength recovery of the upper [11,12,13,14,15] or lower extremities [16,17,18] and improving trunk stability [19,20] in SCI individuals. The most likely mechanisms of tSCS first involve acutely elevating the excitability and plasticity of residual spinal and supraspinal neuronal networks, through the recruitment of afferent fibers from the posterior roots of the spinal cord [11,19,20,21,22,23], and, secondly, and secondly, with training shapes to the more plastic network connectivity toward a more normal coordinated functional state guided via use-dependent mechanism [11,12,13,14,15].

Most tSCS studies so far have focused on lower [16,17,18] or upper extremity function [11,12,13,14,15], addressing both the proof of concept and elaborating on the mechanisms involved. However, much less is known about the feasibility of neuromodulating the brain-to-spinal-cord connectome controlling respiratory function. Mechanisms underlying the action of tSCS combined with physical training, although partially overlapping, may involve different and perhaps synergistic processes leading to a more efficient reorganization of neural circuits [11,22]. A recent report suggested a positive impact of tSCS over the cervical spinal cord on the breathing and coughing ability in a patient with chronic tetraplegia [24]. However, this was a single case study and further research is needed to establish the safety and efficacy of tSCS for improving respiratory function in larger patient populations [24].

Here, our study aimed to investigate the effects of tSCS combined with respiratory training on respiratory parameters in individuals with cervical SCI. The study sought to provide empirical evidence for this hypothesis by evaluating the impact of tSCS on inspiratory and expiratory muscle strength, pulmonary capacity, and subjective perception of respiratory function. The intention was to determine whether the combination of tSCS and respiratory training could lead to improvements in respiratory function more than respiratory training alone.

## 2. Materials and Methods

### 2.1. Subjects

Inclusion criteria were: (i) age between 18 and 70 years; (ii) SCI at the cervical level; (iii) time following SCI > 3 months because stabilization of respiratory function occurs during the first 3 months [25]; (iv) sensation of breathlessness/dyspnea scoring ≥ 3 according to a numerical rating scale (NRS; 0 = no changes, 10 = severe problem) compared to pre-SCI; (v) no change in medical treatment at least one week before and during the study; (vi) agreement to participate after signing a written informed consent form. 

Exclusion criteria were: (i) respiratory problems before SCI; (ii) mechanically ventilated; (iii) unstable SCI or other health condition (cancer, unstable pulmonary or heart disease, etc.); (iv) contraindication for tSCS (e.g., pacemaker); (v) electrical stimulation intolerance; (vi) pregnancy.

SCI subjects were recruited from in-patients at Guttmann Institute (Badalona, Spain), where the study was conducted. The protocol was approved by the Ethics Committee of the Guttmann Institute and was carried out in accordance with the standards of the Declaration of Helsinki (protocol code 2019_317 and date of approval: 4 December 2019).

### 2.2. Clinical Assessment of Spinal Cord Injury

At baseline, the severity and level of SCI was assessed according to the American Spinal Cord Injury Association (ASIA) Impairment Scale (AIS) and the International Standards for Neurological Classification of Spinal Cord, which consists of 5 grades (A, B, C, D, E) [26]. We also collected height and weight of each subject. 

Clinical assessment of SCI, weight, height, and neurophysiology was obtained at baseline only. 

### 2.3. Assessment of Respiratory Function

The study used both subjective and objective measures to evaluate the effectiveness of the intervention. The subjective degree of breathlessness/dyspnea and hypophonia was scored by the subjects using an NRS (0 = no changes, 10 = severe problem) compared to pre-SCI. Objective measures included maximal inspiratory pressure (MIP), maximal expiratory pressure (MEP), and spirometric measures such as forced vital capacity (FVC), forced expiratory volume in 1 s (FEV1), peak expiratory flow (PEF), and forced expiratory flow (FEF25%, FEF50%, FEF75%). These measures are commonly used to assess respiratory motor function and to provide information on respiratory muscle weakness and function. MIP and MEP were measured using the MicroRPM Respiratory Pressure Meter (Micro Direct, Inc., Lewiston, ID, USA), while spirometric measures were performed using the Datospir Micro (Sibelmed; Barcelona, Spain). FVC is the volume of air that can forcibly be blown out after full inspiration; FEV1 is the volume of air that can forcibly be blown out in first 1 s after full inspiration; FEF is the flow (or speed) of air coming out of the lung during the middle portion of a forced expiration, and the usual discrete intervals are 25%, 50%, and 75%; PEF is the maximal flow (or speed) achieved during the maximally forced expiration initiated at full inspiration.

The measurements were conducted by the same physician (HK) for all subjects. The participants were seated, and any belts were loosened. They were also instructed to wear nose clips. The maneuver was demonstrated, and the participants were instructed to inhale or exhale completely (MIP/MEP) or “blast” the air out for 1 s (spirometric measures) following maximal inhalation. Participants were given the opportunity to practice before the final recordings were taken.

All subjects had their MIP, MEP, and spirometric measures recorded three times at baseline (pre-intervention) and three times after the last session (post-intervention).

### 2.4. Neurophysiological Assessment

In our previous research, we conducted a study that demonstrated the effectiveness of using 90% of the rest motor threshold (RT) of the hand muscle for transcutaneous spinal cord stimulation (tSCS) in modulating spinal cord and cortex excitability of upper extremity in healthy subjects. Building upon these findings, our aim in the current study was to apply a similar approach by utilizing 90% of the RT of the diaphragm for tSCS at the C3-4 spinal segment and 90% of the RT of the rectus abdominis muscle for tSCS at the Th9-10. In 4 young SCI subjects (age range: 18–23 years), in order to record the spinal motor response of the diaphragm, we placed the superficial electrodes in the parasternal line, and the monophasic rectangular 1 ms single pulses were delivered at the C3-4 level. For recording the spinal motor response of the rectus abdominis, the electrodes were placed at the umbilical level, and the monophasic rectangular 1 ms single pulses were delivered at the Th9-10 level. The monophasic rectangular 1 ms single pulses were delivered at C3-C4 and at Th9-Th10 using 2 cm diameter hydrogel adhesive electrodes as cathodes (Axelgaard, ValuTrode^®^ Cloth) and two 5 × 12 cm^2^ rectangular electrodes (Axelgaard, ValuTrode^®^ Cloth) placed symmetrically over the iliac crests as anodes [22,23]. Intensity was applied up to 120 mA and RT was defined as the lowest intensity that elicited a spinal motor response ≥50 μV peak-to-peak amplitude. For recordings, we used routine electrodiagnostic equipment (Medelec Synergy, Oxford Instruments; Surrey, UK). Single electric stimulation was applied using the transcutaneous electrical stimulator BioStim-5 (Cosyma Inc., Moscow, Russia). However, we could not obtain any muscle response in diaphragm, nor in the rectus abdominal muscle. Finally, we decided to use biphasic rectangular 1 ms single pulses at C3-4 and C6-7 to record the RT of the abductor pollicis brevis (APB) muscle. These recorded 90% of RT values were then used as the intensity for tSCS at C3-4 and Th9-10, respectively. By using biphasic rectangular 1 ms pulses, we aimed to reduce the discomfort associated with the stimulation and ensure a more tolerable experience for the participants during the threshold determination and stimulation process, because the monophasic pulses were more painful.

### 2.5. Experimental Design 

Initially, the experimental design was a randomized controlled trial that included two groups: (i) tSCS + IMT group, which was tSCS combined with IMT, and (ii) group of IMT alone (control group). Randomization was conducted using a computer-generated list to assign participants to their respective groups. Subjects who underwent IMT alone were given the opportunity to receive tSCS + IMT after completing the initial IMT (control) trial. This allowance was made in response to their request for tSCS, and it allowed them to experience the tSCS + IMT.

Each group underwent a total of five sessions, each lasting approximately 30 min, over the course of one week. Participant preparation for each session took approximately 15–25 min. 

### 2.6. Interventions

Inspiratory muscle training (IMT): We used inspiratory muscle training (IMT) because this was feasible, safe, and a low-cost intervention that may be effective for cSCI subjects, either when performed alone or in conjunction with expiratory muscle training, then known as respiratory muscle training [27].

All patients received IMT using the Fruugo/Australia Lung Trainer 3 Chamber Breathing apparatus for approximately 30 min, five times per week. The inspiratory load was set at 30% of the mean of three trials of MIP. The training began at the smallest inspiratory pressure setting, and once subjects were able to perform the training sessions easily at a particular resistance setting, they progressed to 10 consecutive inspirations at 30% of MIP with a comfortable respiratory velocity, followed by 60 s of resting time, with a total of 15 repetitions (equivalent to 150 inspirations at 30% of MIP).

### 2.7. Transcutaneous Spinal Cord Stimulation

tSCS was applied at the respiratory muscle levels: at C3-4 to promote motor function of the diaphragm and intercostal muscles and at Th9-10 to promote motor function of the abdominal muscles. This approach could hold potential benefits, even if the diaphragm was not directly affected by SCI, because, in healthy individuals, tSCS has the potential to enhance the electrical activity and responsiveness of the spinal cord and brain areas involved in motor control, leading to improvements in motor function and potentially enhancing overall motor control, including in healthy subjects [22,23]. 

tSCS was carried out with a “BioStim-5” (Cosyma Inc., Moscow, Russia). Stimulation was delivered at C3-C4 and at Th9-Th10 using 2 cm diameter hydrogel adhesive electrodes as cathodes (Axelgaard, ValuTrode^®^ Cloth) and two 5 × 12 cm^2^ rectangular electrodes (Axelgaard, ValuTrode^®^ Cloth) placed symmetrically over the iliac crests as anodes [22,23]. tSCS was delivered using biphasic rectangular 1 ms pulses at a frequency of 30 Hz, at 90% of rest motor threshold of APB with each pulse filled with a carrier frequency of 10 kHz. 

tSCS was combined with IMT (as described above) and was turned on during training and turned off during resting periods. 

Clinical, respiratory, and neurophysiological assessments were performed by a physician (HK), the treatment interventions were realized by a physiotherapist (LG). All assessments and interventions were conducted in the morning before 1:00 pm, at the Neurorehabilitation Hospital of the Guttmann Institute. The study took place between January 2020 and January 2023.

### 2.8. Data and Statistical Analysis 

Data were collected for each subject after their respective assessment, and data analysis was performed after completing the assessment of the last subject. Means of MIP, MEP, and spirometric measures were calculated from three repetitions each. Group data were presented as mean ± standard deviation (SD) from individual means for each group separately. Data distribution was examined using the Kolmogorov–Smirnov test. To evaluate the response to the intervention in the two groups, we performed a repeated measure ANOVA on outcome variables, considering the variable “Time” (pre- and post-intervention) as the within-subject factor and the variable “Intervention” (IMT and tSCS + IMT) as the between-subject factor. For parametric data, paired t-tests were used to compare the data between pre- and post-intervention. When the data distribution was not normal (FVC), the Wilcoxon t-test was used instead. Statistical analyses were conducted using a commercial software package (IBM SPSS, version 21.0, SPSS Inc., Chicago, IL, USA). The significance level was set at *p* <0.05. 

## 3. Results

### 3.1. Subjects Clinical and Demographic Characteristics

Thirty-one patients with cervical SCI were recruited to participate in this study. Sixteen patients completed the inclusion criteria and signed a consent form. They were randomly allocated to two experimental groups: IMT alone (n = 11) and tSCS + IMT (n = 5 plus 6 patients who first completed IMT alone and then, after at least one week of break, received tSCS + IMT as shown in flow diagram in Figure 1). This approach was implemented to prevent any potential sustained effects of IMT from influencing the outcomes when the tSCS + IMT was introduced (Figure 1).

The mean age was 29.3 ± 10.1 years for tSCS + IMT and 37.0 ± 5.9 years for the IMT group. All subjects were male except one female in the tSCS + IMC group. The mean time since SCI was 8.1 ± 1.8 months for tSCS + IMT and 7.5 ± 1.7 for the IMT group. Lesion levels of SCI were C3/C4/C5/C6/C7, Lesion level of SCI was similar in both groups (at C3/C4/C5/C6/C7 in IMT group: n = 1/5/3/0/2; tSCS+IMT group: n = 1/5/2/1/2 respectively (Table 1). The severity of SCI according to AIS A/B/C/D was distributed in the IMT group 2/4/4/1, and in the tSCS + IMT group 2/4/4/1 (Table 1). The two groups were comparable in terms of age, time since SCI, lesion level, and SCI severity as assessed by the AIS scale. The statistical analysis showed no significant differences between the groups for these variables (*p* > 0.05 for each comparison, Table 1).

All data (breathlessness/dyspnea, hypophonia, MIP, MEP, spirometric measures) were similar between both groups (*p* > 0.05) at baseline (pre-treatment) (Table 2).

### 3.2. Respiratory Assessment 

#### 3.2.1. Subjective Evaluation

Subjects reported significant improvement in breathlessness/dyspnea post intervention (F = 8.272, *p* < **0.009**; η^2^ = 0.293) with a significant interaction of Time × Intervention (F = 19.449, *p* < **0.001)** (Table 2; Figure 2). Breathlessness/dyspnea improved significantly compared to baseline in the tSCS + IMT group (*p* = 0.002, paired *t*-test) but not in the IMT group (*p* = 0.441, Figure 2).

Hypophonia improved significantly (F = 18.06, *p* < **0.001**, η^2^ = 0.475) with a significant interaction of Time × Intervention (F = 9.552, *p* = **0.006,** Table 2). Hypophonia improved in the tSCS + IMT significantly (paired *t*-test, *p* = 0.002) but not in the IMT group (*p* = 0.341, Figure 2).

#### 3.2.2. Objective Evaluation

Inspiratory thorax muscle strength measured by MIP improved significantly compared to baseline (F= 4.452, *p* = 0.048, η^2^ = 0.182), with a significant interaction of Time × Intervention (F = 5.813, *p* = 0.026, Table 2; Figure 3). After the last session, MIP improved significantly with respect to baseline in the tSCS + IMT group (paired *t*-test, *p* = 0.004) but not in the IMT group (*p* = 0.814, Figure 3).

Expiratory thorax muscle strength (MEP) also improved significantly (F = 15.240, *p* < **0.01**; η^2^ = 0.432), with a significant interaction of Time × Intervention (F = 6.708, *p* = **0.017**, Table 2; Figure 3). The improvement was significant in comparison to baseline in the tSCS + IMT group (*p* = 0.004, paired *t*-test) but not in the IMT group (*p* = 0.233, Figure 3).

Spirometric measures: FVC increased significantly compared to baseline following the last session of tSCS + IMT (Wilcoxon test, *p* = 0.013) but did not change significantly in the IMT group (*p* = 0.534) (Table 2, Figure 4). There were no significant changes in the other spirometric measures (FEV_1_, FEV_1_/FVC, PEF, FEF50%, FEV25%/75%, FEV_1_/FEV0.5) in either group (*p* > 0.05, Table 2).

Adverse effects: Seven subjects in the tSCS + IMT group complained of mild to moderate pain (range: 1–5) around the tSCS electrodes, particularly in the cervical segment, but no one left the study. None had significant changes in blood pressure during tSCS.

## 4. Discussion 

This study demonstrates that tSCS combined with IMT improved respiratory function both through self-report measures such as breathlessness and hypophonia and through objective assessments such as inspiratory and expiratory muscle strength and pulmonary vital capacity. It is noteworthy that these improvements were observed after only one week of the combined intervention, indicating the potential for relatively rapid effects. 

In the literature, only one article [24] reported potential therapeutic benefits of tSCS for improving respiratory function in a 39-year-old man with a complete SCI at the C5 level. The subjects received tSCS over the cervical spinal cord for 20 min twice a day in addition to respiratory muscle training, for a total of 8 weeks. The authors used biphasic pulses consisting of a 10 kHz carrier pulse at 30 Hz, applying the cathode electrodes at C3-4, C5-6, or Th1-2, and two anode electrodes over both shoulders. The authors selected the stimulation site with the lowest intensity that generated the greatest functional respiratory response, using inspiratory capacity and forced expiratory volume in 1 s (FEV1) as their indicator. Their results showed significant improvements in MIP, MEP, and FVC, as well as in subjective measures of dyspnea and cough effectiveness [24]. In our study, we used similar characteristic of tSCS (biphasic 10 kHz carrier pulses and at 30 Hz) in 11 cSCI subjects. On the other side, (i) we used tSCS at two spinal segments (C3-4 and Th9-10), (ii) using the 90% of rest motor threshold in APB muscle. (iii) The cathode electrodes were placed at C3-4 and Th9-10, where each one was connected to an anode electrode placed symmetrically over the iliac crests, and (iv) the duration of the tSCS was 5 days. 

Our subjects with cervical SCI did not present severe respiratory problems because we excluded those with mechanically ventilated cervical SCI and included subjects with stabilized respiratory function. However, we applied tSCS at C3-4 to exert a beneficial effect on respiratory parameters in the diaphragm, which was not affected severely in this study because, in our previous study, tSCS could promote the muscle function of the non-affected muscles in healthy subjects [22,23]. The diaphragm provides the major driving force for inspiration, which is innervated by the phrenic nerve, derived primarily from the C3 through C5 lower motor neurons (LMNs), most of which reside within the C4 spinal segment. High-cervical lesions may damage phrenic motor neurons and/or disrupt descending bulbospinal pathways [28]. However, lesions rostral to the C3 or C4 level spare phrenic LMNs but disrupt descending bulbospinal upper motor neuron signal transmission from pattern-generating respiratory centers in the medulla oblongata. Both these injury profiles typically result in a dependence on artificial, external respiratory replacement, traditionally a tracheostomy tube, and mechanical positive pressure ventilation. However, this was not the case in our SCI subjects, because such severe cases were not included in our study.

In this study, combined tSCS with IMT improved respiratory function both through self-report measures and through objective assessments. Additionally, the comparison with IMT alone suggests that tSCS may provide additional benefits beyond IMT alone. It was previously published that tSCS combined with muscle activity demonstrates a greater ability to modulate the muscle force, spinal cord, and cortex excitability compared to using intervention alone in healthy subjects [22]. This suggests that the combined approach may have synergistic effects in promoting neural modulation and plasticity [11,22]. Mechanisms underlying the action of tSCS combined with physical training, although partially overlapping, may involve different and perhaps synergistic processes leading to a more efficient reorganization of neural circuits [11,22,23].

tSCS was applied at two spinal segments: at the C3-4 level to exert a beneficial effect on respiratory parameters at the diaphragm and intercostal muscles and at the Th9-10 tSCS to exert the same through abdominal muscles. The placement of electrodes for tSCS is an important consideration in order to target specific segments of the spinal cord and promote desired functional outcomes. For promoting upper extremity function, the electrodes were placed at the C3-4, C5, and/or C7-8 segments of the spinal cord [11,12,13,14,15]. These segments are in proximity to the innervation of the upper limb muscles and are targeted to enhance the motor control and responsiveness of the upper extremities. Similarly, for promoting lower extremity function, the electrodes were placed at the Th11-12 and/or L1-2 segments of the spinal cord [16,17,18]. These segments are relevant for the innervation of the lower limb muscles and were selected to optimize the stimulation effects on the lower extremities.

The principal mechanism of tSCS is a non-invasive activation of inaccessible neuronal networks of the spinal cord likely including the recruitment of afferent fibers (large–medium) in the posterior root in order to elevate spinal network excitability [20,29]. The excitability of spinal interneuronal networks without directly producing action potentials can be readily modulated by changing the networks’ physiological state [30]. In addition, the recruitment of cutaneous mechanoreceptors surrounding the electrodes may also contribute to the neuromodulatory effects of tSCS through these polysynaptic connections [31]. In animal models, Guiho et al. [32] observed a potentiation of supraspinal evoked responses with both dorsal epidural SCS and tSCS over the C3-4 and C7–T1 intervertebral spaces in monkeys, but facilitation was stronger with dorsal epidural SCS. It has been reported [10,33] that high-frequency (300-Hz) SCS via a single epidural electrode at the second thoracic spinal level (T2) applied to the dorsal epidural was capable of evoking a physiological recruitment pattern of the inspiratory musculature in canine models of SCI. A reliable approach used to promote functional recovery is to neuromodulate the preserved spinal cord connectome innervating the preserved muscles. The underlying hypothesis for the effect of tSCS is a neuromodulation of spinal sensorimotor networks above, within, and below the lesion toward an elevated functional state that enables and amplifies voluntary motor control. Repeated tSCS over multiple treatment sessions may eventually trigger a cascade of adaptive events, ultimately leading to functional neural reorganization [24], and this can be manifested as chronic (adaptive, learned) functions that can persist for minutes to days. This activity-dependent response is also well suited for respiratory network reorganization and functional recovery following spinal cord injury [24].

### Strength and Limitations

This study had a few limitations. (i) We included only individuals with spinal cord injuries with an evolution of more than 3 months. Nevertheless, the IMT group did not show significant changes, whereas tSCS + IMT resulted in significant improvements in both subjective and objective parameters. (ii) We used a subjective scale of breathlessness for the inclusion criteria. Using a subjective scale for the inclusion criteria may indeed have certain limitations, as subjective measures can be influenced by individual perception and interpretation. However, it is encouraging to note that, despite this limitation, the objective assessments still showed significant differences. (iii) Although the time since injury was similar among the participants, individual differences in injury severity or other factors may have influenced the results. (iv) Blinding was not possible, as subjects knew they were receiving electrical stimulation due to the high intensity of tSCS. (v) Long-term follow-up was not possible due to the COVID-19 pandemic and the aim to reduce contact risk. (vi) Finally, the number of participants was small and the stimulation period was short (five days only). Despite these limitations, the study revealed promising results in various respiratory measures.

## 5. Conclusions

Current evidence supports the potential of tSCS as an adjunctive therapy to accelerate and enhance the rehabilitation process for respiratory impairments following SCI. However, further research is needed to validate these results and establish the long-term benefits of tSCS in this population. Additionally, it is important to explore and understand the optimal parameters of tSCS, including intensity, frequency, and stimulation at different segments of the spinal cord. Tailoring the tSCS intervention to individual needs and optimizing the stimulation parameters may contribute to maximizing the therapeutic benefits and improving clinical outcomes for SCI individuals.

## Figures and Tables

**Figure 1 biomedicines-11-02121-f001:**
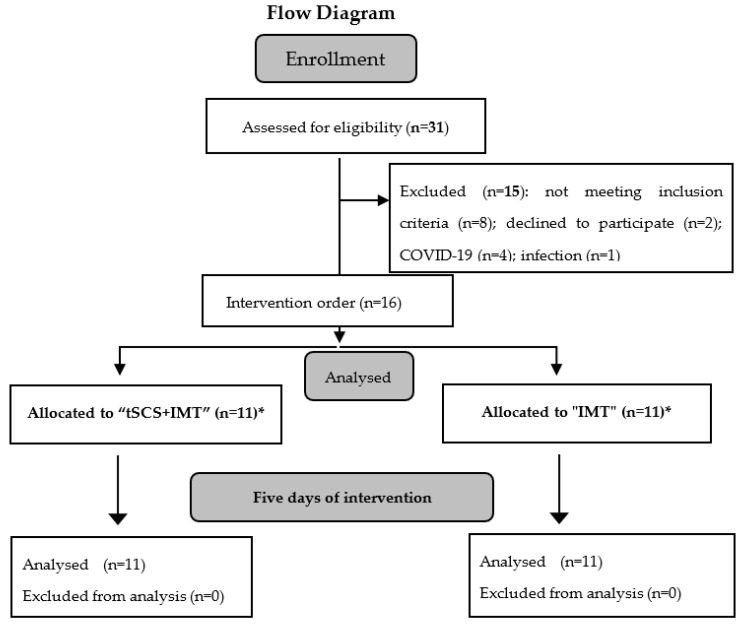
Participant flow throughout the trial duration. IMT: Inspiratory muscle training; tSCS: transcutaneous spinal cord stimulation; *: 6 patients first received IMT alone and, at least one week later, tSCS + IMT.

**Figure 2 biomedicines-11-02121-f002:**
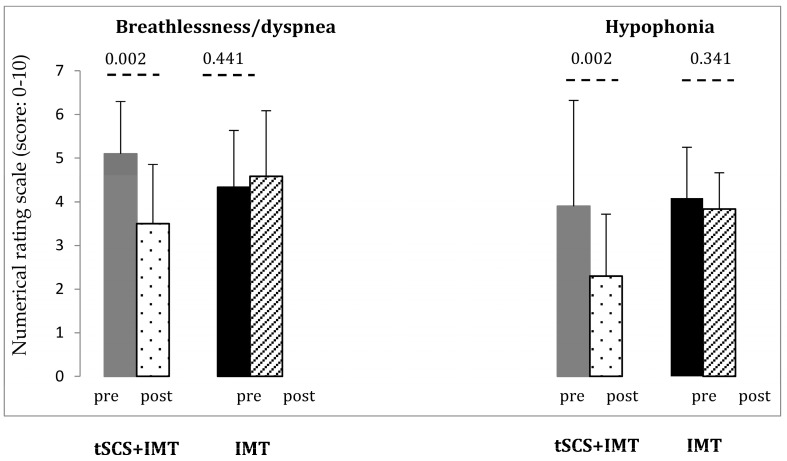
Changes in breathlessness/dyspnea and in hypophonia measured by numerical rating scale (NRS). IMT: Inspiratory muscle training; tSCS: transcutaneous spinal cord stimulation. NRS: 0 = no changes and 10 severe problems following SCI in comparison to pre-SCI. *p*-value according to paired *t*-test.

**Figure 3 biomedicines-11-02121-f003:**
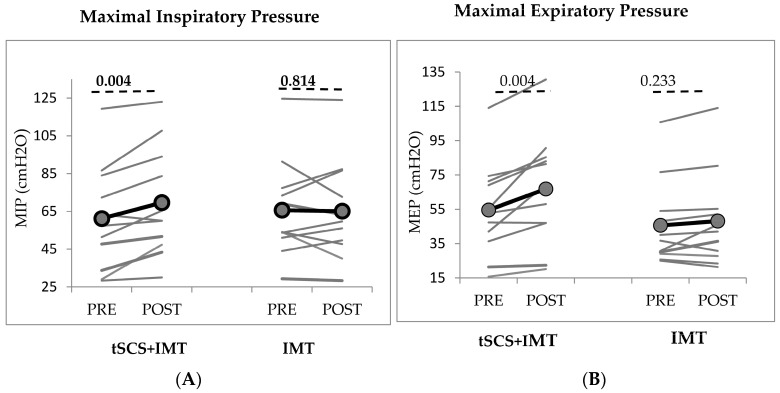
Changes: (**A**) in MIP (maximal inspiratory pressure) and, (**B**) in MEP (maximal expiratory pressure) with tSCS + IMT and with IMT. IMT: inspiratory muscle training; tSCS: transcutaneous spinal cord stimulation. The grey lines represent individual cSCI subjects and the black lines represent the group mean. Significant improvement post-tSCS + IMT in MIP and MEP. *p*-value according to paired *t*-test.

**Figure 4 biomedicines-11-02121-f004:**
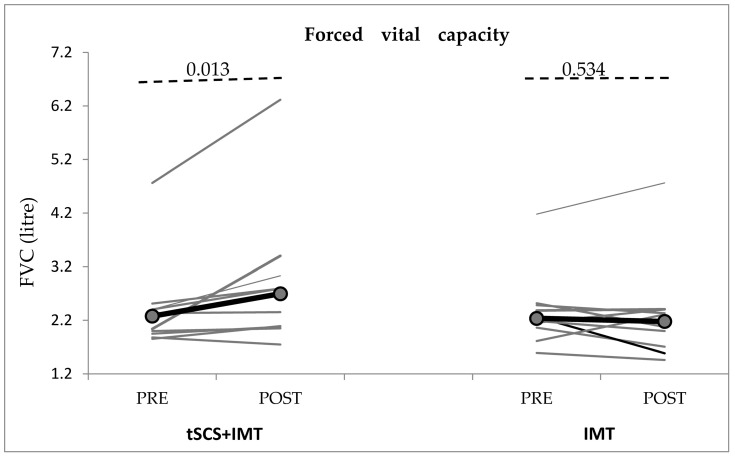
Changes in forced vital capacity (FVC) in both conditions. IMT: inspiratory muscle training; tSCS: transcutaneous spinal cord stimulation. The grey lines represent individual cSCI subjects and the black lines represent the group means. Significant improvement post-tSCS + IMT. *p*-value according to Wilcoxon-*t*-test.

**Table 1 biomedicines-11-02121-t001:** Clinical and demographic characteristics of the subjects with spinal cord injury, heights, weight, and intensity of stimulation at C3-4 (C3) and Th9-10 (Th9) vertebral segments. The last two columns consist of the intensity of stimulation of each of the segment levels that received tSCS.

	Age	Sex	SCI Etiology	SCI Level	AIS	Time Since SCI (Month)	Height (cm)	Weight (kg)	tSCS Intensity at C3 (mA)	tSCS Intensity at Th9 (mA)
IMT	38 *	M	Trauma	C7	B	7	165	54	-	-
IMT	36	M	Trauma	C4	B	9	172	57	-	-
IMT	36	M	Trauma	C4	C	10	167	61	-	-
IMT	31	M	Trauma	C5	A	8	170	76	-	-
IMT	18 *	M	Trauma	C4	B	8	174	48	-	-
IMT	45	M	Trauma	C4	C	6	175	62	-	-
IMT	21 *	M	Trauma	C5	B	9	176	60	-	-
IMT	18 *	M	Trauma	C3	A	7	179	58	-	-
IMT	23 *	F	Trauma	C7	C	6	170	54	-	-
IMT	47	M	Trauma	C4	D	9	172	83.2	-	-
IMT	26 *	M	Trauma	C5	C	4	178	82.5	-	-
tSCS + IMT	25	M	Trauma	C4	A	9	186	76	67	60
tSCS + IMT	46	M	Trauma	C4	C	8	174	76	80	80
tSCS + IMT	28	M	Trauma	C6	A	7	176	71	77	90
tSCS + IMT	18 *	M	Trauma	C4	B	9	174	48	44	52
tSCS + IMT	35	M	Trauma	C4	C	7	175	62	34	40
tSCS + IMT	21 *	M	Trauma	C5	B	9	176	60	52	66
tSCS + IMT	18 *	M	Trauma	C3	B	10	179	58.5	66	70
tSCS + IMT	23 *	F	Trauma	C7	C	9	170	54	62	62
tSCS + IMT	26 *	M	Trauma	C5	C	5	178	82.5	57	61
tSCS + IMT	34	M	Trauma	C4	D	4	175	68.5	49	59
tSCS + IMT	38 *	M	Trauma	C7	B	8	165	54	78	82
*p*	0.08					1.00	0.20	0.79		

* SCI subjects who first received IMT and, at least one week later, tSCS + IMT. M: male; C: cervical; Th: thoracic.

**Table 2 biomedicines-11-02121-t002:** The data of subjective and objective assessment of respiratory changes.

		PRE	POST	*p* **
tSCS + IMT	breathlessness	5.09 ± 1.14	3.36 ± 1.36	F = 8.272, *p* < **0.009**, η^2^ = 0.293 TimexIntervention F = 19.449, *p* < **0.001** η^2^ = 0.493
IMT		4.91 ± 1.67	4.91 ± 1.64	
*p* *		0.336		
tSCS + IMT	hypophonia	4.00 ± 2.32	2.27 ± 1.35	F = 18.06, *p* < **0.001**, η^2^ = 0.475 TimexIntervention F = 9.552, *p* = **0.006**; η^2^ = 0.323
IMT		4.18 ± 1.17	3.91 ± 0.83	
*p* *		0.819		
tSCS + IMT	MIP	61.18 ± 28.01	69.64 ± 28.93	F = 4.452, *p* < **0.048**, η^2^ = 0.182 TimexIntervention F = 5.813, *p* = **0.026**; η^2^ = 0.225
IMT		65.61 ± 26.01	65.03 ± 26.69	
*p* *		0.705		
tSCS + IMT	MEP	54.48 ± 27.53	66.80 ± 32.41	F = 15.240, *p* < **0.01**, η^2^ = 0.432 TimexIntervention F = 6.708, *p* = **0.017**; η^2^ = 0.251
IMT		45.58 ± 25.15	48.09 ± 27.71	
*p* *		0.439		
spirometric measures				
		**PRE**	**POST**	
tSCS + IMT	FVC (L)	2.28 ± 0.93	2.70± 1.37	*p*^&^ = **0.013** *p* ^&^ = 0.534
IMT		2.24 ± 0.79	2.18 ± 0.98	
*p* ^@^		0.898		
tSCS + IMT	FEV_1_ (L)	1.58 ± 0.57	1.75 ± 0.70	F = 0.067, *p* = 0.799, η^2^ = 0.003 TimexIntervention F = 6.708, *p* = **0.018**; η^2^ = 0.251
IMT		1.78 ± 0.46	1.57 ± 0.49	
*p* *		0.406		
tSCS + IMT	FEV_1_/FVC (%)	72.30 ± 22.37	70.21 ± 20.92	F = 2.196, *p* = 0.154, η^2^ = 0. 099 TimexIntervention F = 0.062, *p* = 0.806; η^2^ = 0.003
IMT		78.85 ± 12.51	75.92 ± 13.95	
*p* *		0.402		
tSCS + IMT	PEF (L/s)	2.90 ± 1.39	3.04 ± 1.32	F = 0.004, *p* = 0.950, η^2^ = 0.000 TimexIntervention F = 0.840, *p* = 0.371; η^2^ = 0.042
IMT		2.94 ± 0.73	2.82 ± 0.85	
*p* *		0.772		
tSCS + IMT	FEF50% (L/s)	1.57 ± 0.68	1.58 ± 0.70	F = 0.232, *p* = 0.636, η^2^ = 0.012 TimexIntervention F = 0.371, *p* = 0.550; η^2^ = 0.019
IMT		1.78 ± 0.49	1.67 ± 0.67	
*p* *		0.482		
tSCS + IMT	FEF25%/75% (L/s)	1.62 ± 0.83	2.46 ± 3.28	F = 0.851, *p* = 0.3670, η^2^ = 0. 041 TimexIntervention F = 2.388, *p* = 0.268; η^2^ = 0.061
IMT		1.66 ± 0.46	1.57 ± 0.61	
*p* *		0.639		
tSCS + IMT	FEV_1_/FEV0.5	1.44 ± 0.18	1.44 ± 0.18	F = 0.30, *p* = 0.864; η^2^ = 0.001 TimexIntervention F = 0.74, *p* = 0.788, η^2^ = 0. 004
IMT		1.45 ± 0.15	1.45 ± 0.17	
*p* *		0.880		

* *p*. the differences between tSCS + IMT vs. IMT group at baseline condition according to *t*-test except FVC; ^@^: Mann–Whitney-U test; **: *p*. Repeated measure ANOVA between pre vs. post of tSCS + IMT vs. IMT group; except FVC, ^&^
*p* value according to Wilcoxon-*t*-test. η^2^ = effect size (η^2^ = 0.01: small effect; η^2^ = 0.06: medium effect; η^2^ = 0.14: large effect); Maximum inspiratory and expiratory muscle pressure: MIP and MEP, respectively; FVC: forced vital capacity (FVC); FEV_1_: forced expiratory volume in 1 s; PEF: peak expiratory flow; FEF: forced expiratory flow.

## Data Availability

The data presented in this study are available on request from the corresponding author.

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
