# Peer review of "Transcutaneous Spinal Cord Stimulation Improves Respiratory Muscle Strength and Function in Subjects with Cervical Spinal Cord Injury: Original Research"

_biomedicines, 2023, doi:10.3390/biomedicines11082121_

Round 1

Reviewer 1 Report

In this study the authors investigated if transcutaneous spinal cord stimulation applied at the cervical and thoracic levels can improve respiratory function in cSCI subjects. This study was a randomized controlled trial. However, it is not clear for readers if this study had been registered at ClinicalTrials,gov. This is a major concern and should not be sent out for peer review.

In addition, the three-line table should be used in the manuscript.

fine

Author Response

We thank for the constructive comments of the reviewer. 

In this study the authors investigated if transcutaneous spinal cord stimulation applied at the cervical and thoracic levels can improve respiratory function in cSCI subjects. This study was a randomized controlled trial. However, it is not clear for readers if this study had been registered at ClinicalTrials,gov. This is a major concern and should not be sent out for peer review.

Answer: This study was not registered atClinicalTrials,gov.

While registering clinical trials on platforms like ClinicalTrials.gov is encouraged for transparency and to promote research integrity, we understand that logistical and resource constraints can make it challenging to meet these requirements.

It is important to note that the absence of trial registration does not diminish the value or significance of the study itself. The results and findings from our study can still contribute to the existing knowledge and understanding of the topic.

In addition, the three-line table should be used in the manuscript.

Answer : Done as suggested.

Reviewer 2 Report

This was an interesting study where the authors investigated the effect of non-invasive transcutaneous spinal cord stimulation (tSCS) on respiratory function improvement in patients. While the premise of the study is interesting, the manuscript needs major revision and editing (it appears as if no one really reviewed the pdf before sending it out for review) so that it will be appropriate for the peer-reviewed process.

General comments:

1.     Authors have hypothesized in combining IMT (inspiratory muscle training) with tSCS at the C3-4 level may exert a beneficial effect on respiratory parameters at the diaphragm and intercostal muscles and Th9-10 tSCS do the same through abdominal muscles. What is the proposed mechanism of action for cervical and thoracic tSCS? This should be included in the discussion as this will be important for understanding the results of this study as well as future research direction.

2.     Has there been a validation of this hypothesis in a preclinical model? It is better to characterize the neuronal networks and pathways involved in mediating the effects of tSCS before attempting into the clinical setting.

3.     It seems the authors didn’t assess subjects for any side-effects after tSCS. Despite the promising features of tSCS compared to epidural stimulation, it is associated with a certain spatiotemporal limitation relative to the accuracy and specificity of the current distribution. Undesired stimulation can coactivate functionally unrelated but anatomically adjacent neural pathways. Unspecific coactivation of nontargeted motor and/or sensory neural structures can thus evoke adverse stimulation effects, such as muscle twitches, jerks, spasms, dysesthesias, pain, or even autonomic dysreflexia, particularly in individuals with incomplete SCI. (Hachmann, J. T., Grahn, P. J., Calvert, J. S., Drubach, D. I., Lee, K. H., & Lavrov, I. A. (2017, September). Electrical neuromodulation of the respiratory system after spinal cord injury. In Mayo Clinic Proceedings (Vol. 92, No. 9, pp. 1401-1414). Elsevier.

4.     Authors induced tSCS over C3-4 or Th9-10, however, they didn’t mention the individual effect of cervical or thoracic tSCS. Were there any differences in the effect of cervical versus thoracic tSCS on respiratory function?

5.     The inclusion criteria were SCI at the cervical level. However, mostly rostral cervical SCI (C3-C5) results in more pronounced respiratory dysfunction. So, every patient with cervical SCI may not be the ideal candidate to study the effect of tSCS on respiration. These variations should be noted in the discussion section.

o   Note: High-cervical lesions may damage phrenic motor neurons and/or disrupt descending bulbospinal pathways. The diaphragm provides the major driving force for inspiration, which is innervated by the phrenic nerve, derived primarily from the C3 through C5 lower motor neurons (LMNs), most of which reside within the C4 spinal segment. Two profiles of SCI can cause diaphragmatic paralysis and complete respiratory failure: injuries at the mid-cervical region, which directly damage or destroy phrenic LMNs, and lesions rostral to the C3 or C4 level, which spare phrenic LMNs, but disrupt descending bulbospinal upper motor neuron signal transmission from pattern-generating respiratory centers in the medulla oblongata. (Hachmann, J. T., Grahn, P. J., Calvert, J. S., Drubach, D. I., Lee, K. H., & Lavrov, I. A. (2017, September). Electrical neuromodulation of the respiratory system after spinal cord injury. In Mayo Clinic Proceedings (Vol. 92, No. 9, pp. 1401-1414). Elsevier.)

6. IMT proposed to correlate respiration after SCI by improving respiratory muscle strength and fatigue tolerance. It is surprising that IMT did not result in significant improvements in objective and subjective measurements e.g., spirometric, NRS for breathlessness/dyspnoea? Please see the below paper for the effect of IMT. Please add this to the discussion.

o   Goosey-Tolfrey, V., Foden, E., Perret, C., & Degens, H. (2010). Effects of inspiratory muscle training on respiratory function and repetitive sprint performance in wheelchair basketball players. British journal of sports medicine, 44(9), 665-668.

o   West, C. R., Taylor, B. J., Campbell, I. G., & Romer, L. M. (2014). Effects of inspiratory muscle training on exercise responses in Paralympic athletes with cervical spinal cord injury. Scandinavian journal of medicine & science in sports, 24(5), 764-772.

o   McDonald, T., & Stiller, K. (2019). Inspiratory muscle training is feasible and safe for patients with acute spinal cord injury. The journal of spinal cord medicine, 42(2), 220-227.

7. Why authors used a subjective scale Numerical Rating Scale (NRS) for the inclusion criteria and not an objective scale e.g., MEP, MIP, FEV1, PEF, FVC, and FEF? Those would be more accurate and systematic.

8. It is recommended to have a sham group, implanting tSCS electrodes on the subjects without stimulation, to evaluate the effect of tSCS. Although, it would be hard to have blind subject, examiner and research staff should be blinded about sham and actual the treatment combination.

9. As a suggestion, it would have been better to measure the effect of different frequencies and lower intensities of tSCS on the improvement of respiratory function (the excitation of phrenic interneurons/bulbospinal neurons). Lower intensity may also show beneficial effects and patient tolerance while reducing the probable side effects. Were there any side effects noted by the patients? Please add this information and discussion point about choosing the high intensity stimulation.

o   DiMarco, A. F., & Kowalski, K. E. (2013). Spinal pathways mediating phrenic activation during high-frequency spinal cord stimulation. Respiratory physiology & neurobiology, 186(1), 1-6.

o   DiMarco, A. F., Kowalski, K. E., Geertman, R. T., & Hromyak, D. R. (2009). Lower thoracic spinal cord stimulation to restore cough in patients with spinal cord injury: results of a National Institutes of Health–sponsored clinical trial. Part I: methodology and effectiveness of expiratory muscle activation. Archives of physical medicine and rehabilitation, 90(5), 717-725.

o    

10. It is important to note that the only differences detected were in measurements that were self-reported (The subjects evaluated their sensation of breathlessness/dyspnea and hypophonia compared to pre-SCI using a numerical rating scale.) and no significant effects of the treatment was seen in any of the measured metrics. This should be discussed.  

Specific Comments

1.     Abstract, the acronym “FVC” should be spelled out.

2.     Line 137; the authors mentioned “young” patients. Was there a classification of young and old patients?

3.     There were several spots where “dyspnea” was misspelled. Please find them and edit them appropriately.

4.     Why authors used a low threshold NRS (≥ 3) for scoring the sensation of breathlessness/dyspnea for the inclusion criteria? (Subjective scale)

5.     Stimulation location: authors described the position of the electrodes (cathode and anode) for the first time in the discussion section (page 11 line 302). It’s recommended to report this information in the Materials and Methods section.

6.     Did the authors use  single electrical stimulation, or do you mean single pulse? (page 3 line 137)? I thought it was biphasic (according to line 308).

7.     What was the pulse width used for each 10K carrier for tSCS?

8.     Based on the text it’s unclear whether the authors used biphasic or monophasic for tSCS? On page 3 line 137 they mentioned monophasic, while on page 4 line 185 they reported using biphasic for tSCS.

9.     If authors used monophasic for tSCS, what was the reason for that? And was it positive or negative monophasic?

10.  In addition, did the authors use a single pulse (page 3 line 137) or 10K carrier (page 4 line 186) stimulation for tSCS?

11.  Where authors placed/implanted recording electrodes?

12.  Why did the authors not evaluate tSCS alone? It could be more meaningful to compare tSCS and IMT individually. Please add to the discussion.

13.  Fig 1. There is nothing in some of the boxes in this figure. Please review and edit!

14.  Fig 2. This figure is very hard to read.  There are also misplaced texts in the figure that is on top of other texts making it impossible to read. Please edit. Figure legend for Fig 2 is also incomprehensible. The different bars are also not labelled. Please edit. P values should be in the text in the legend. It is hard to decipher what the p values are from if it is floating around in the figure. Y axis labels are unclear!

15.  Fig 3. The legend states “A” and “B” but there is no “A” or “B” in the figure. What does “Max” refer to in the figure? Again, the p values are floating all around the figure and it is unclear what those numbers are associated with. Please edit. Y axis labels are unclear or not present!

16.  Fig 4. The same problems as seen in Figures 2 & 3. Please edit the figure and figure legend thoroughly so that they can be legible.

17.  Change “his” (page 11 line 321) to “this”.

18.  There were multiple places in the manuscript with extra spaces. Please review and edit.

see above 

Author Response

We thank to reviewer for her/his constructive comments.

  1. Authors have hypothesized in combining IMT (inspiratory muscle training) with tSCS at the C3-4 level may exert a beneficial effect on respiratory parameters at the diaphragm and intercostal muscles and Th9-10 tSCS do the same through abdominal muscles. What is the proposed mechanism of action for cervical and thoracic tSCS? This should be included in the discussion as this will be important for understanding the results of this study as well as future research direction.

   Answer: The placement of electrodes for tSCS is an important consideration in order to target specific segments of the spinal cord and promote desired functional outcomes. In the context of your study, the electrodes were placed at specific levels to target the upper extremity and lower extremity. For promoting upper extremity function, the electrodes were placed at the C3-4, C5, and C7-8 segments of the spinal cord. Similarly, for promoting lower extremity function, the electrodes were placed at the Th11-12 and L2 segments of the spinal cord. These segments are in proximity to the innervation of the upper or lower limb muscles and are targeted to enhance the motor control and responsiveness

We added additional information in the discussion about the proposed mechanism of action for cervical and thoracic tSCS.

  1. Has there been a validation of this hypothesis in a preclinical model? It is better to characterize the neuronal networks and pathways involved in mediating the effects of tSCS before attempting into the clinical setting.

  Answer: In the discussion we added mor information about animal studies.  "In animal models, Guiho et al (32) observed potentiation of supraspinal evoked responses with both dorsal epidural SCS and tSCS over the C3–4 and C7–T1intervertebral spaces in monkeys, but facilitation was stronger with dorsal epidural SCS. DiMarco, Kowalski, (33,34)  reported that high-frequency (300-Hz) SCS via a single epidural electrode at the second thoracic spinal level (T2) applied to the dorsal epidural was capable of evoking a physiological recruitment pattern of the inspiratory musculature in canine models of SCI."

  1. It seems the authors didn’t assess subjects for any side-effects after tSCS. Despite the promising features of tSCS compared to epidural stimulation, it is associated with a certain spatiotemporal limitation relative to the accuracy and specificity of the current distribution. Undesired stimulation can coactivate functionally unrelated but anatomically adjacent neural pathways. Unspecific coactivation of nontargeted motor and/or sensory neural structures can thus evoke adverse stimulation effects, such as muscle twitches, jerks, spasms, dysesthesias, pain, or even autonomic dysreflexia, particularly in individuals with incomplete SCI. (Hachmann, J. T., Grahn, P. J., Calvert, J. S., Drubach, D. I., Lee, K. H., & Lavrov, I. A. (2017, September). Electrical neuromodulation of the respiratory system after spinal cord injury. In Mayo Clinic Proceedings (Vol. 92, No. 9, pp. 1401-1414). Elsevier.

   Answer: We are sorry for this missing data. In all patients and we asked the side effect of tSCS and recorded the blood pressure during tSCS. Now we added this information in the result section: "Advers effects: Seven subjects in the tSCS+IMT group complained of mild to moderate pain (range: 1-5) around the tSCS electrodes, particularly in the cervical segment, but no one left the study. The blood pressure was recorded during tSCS, and  nobody had dysreflexia."

  1. Authors induced tSCS over C3-4 or Th9-10, however, they didn’t mention the individual effect of cervical or thoracic tSCS. Were there any differences in the effect of cervical versus thoracic tSCS on respiratory function?

Answer: This is an important point that requires further investigation. In our study, we built upon previous experiences with two-segment transcutaneous spinal cord stimulation (tSCS) in both healthy individuals and those with SCI. We observed improvements in clinical and neurophysiological assessments for upper extremities  following this approach. This highlights the need for additional studies to compare different tSCS approaches, including single-segment, two-segment, and potentially even more extensive stimulation strategies. These studies can further explore the optimal parameters and configurations for tSCS to maximize its therapeutic benefits.

Overall, while our study contributes to the growing body of knowledge on the potential benefits of tSCS, further investigations are necessary to determine the most effective stimulation protocols and to better understand the underlying mechanisms of action. By continuing to explore and refine tSCS techniques, we can advance the field and ultimately improve outcomes for individuals with spinal cord injury.

Kumru, H.; et al. Cervical Electrical Neuromodulation Effectively Enhances Hand Motor Output in Healthy Subjects by Engaging a Use-Dependent Intervention. J. Clin. Med. 2021; 

Kumru H, Transcutaneous Electrical Neuromodulation of the Cervical Spinal Cord Depends Both on the Stimulation Intensity and the Degree of Voluntary Activity for Training. A Pilot Study J Clin Med. 2021;

García-Alén, L, et al.Transcutaneous Cervical Spinal Cord Stimulation Combined with Robotic Exoskeleton Rehabilitation for the Upper Limbs in Subjects with Cervical SCI: Clinical Trial. Biomedicines. 2023.

  1. The inclusion criteria were SCI at the cervical level. However, mostly rostral cervical SCI (C3-C5) results in more pronounced respiratory dysfunction. So, every patient with cervical SCI may not be the ideal candidate to study the effect of tSCS on respiration. These variations should be noted in the discussion section.

o   Note: High-cervical lesions may damage phrenic motor neurons and/or disrupt descending bulbospinal pathways. The diaphragm provides the major driving force for inspiration, which is innervated by the phrenic nerve, derived primarily from the C3 through C5 lower motor neurons (LMNs), most of which reside within the C4 spinal segment. Two profiles of SCI can cause diaphragmatic paralysis and complete respiratory failure: injuries at the mid-cervical region, which directly damage or destroy phrenic LMNs, and lesions rostral to the C3 or C4 level, which spare phrenic LMNs, but disrupt descending bulbospinal upper motor neuron signal transmission from pattern-generating respiratory centers in the medulla oblongata. (Hachmann, J. T., Grahn, P. J., Calvert, J. S., Drubach, D. I., Lee, K. H., & Lavrov, I. A. (2017, September). Electrical neuromodulation of the respiratory system after spinal cord injury. In Mayo Clinic Proceedings (Vol. 92, No. 9, pp. 1401-1414). Elsevier.)

  Answer: Thank you for bringing this point to our attention, and we will ensure to adequately address it in the final manuscript.

On the other side, we acknowledge that the inclusion criteria for our study focused on individuals with spinal cord injury (SCI) who were able to breathe spontaneously but experienced subjective sensations of respiratory difficulties (L.85-87). We specifically excluded individuals who required mechanical ventilation.

We added this information suggested by reviewer in the discussion.

  1. IMT proposed to correlate respiration after SCI by improving respiratory muscle strength and fatigue tolerance. It is surprising that IMT did not result in significant improvements in objective and subjective measurements e.g., spirometric, NRS for breathlessness/dyspnoea? Please see the below paper for the effect of IMT. Please add this to the discussion.

o   Goosey-Tolfrey, V., Foden, E., Perret, C., & Degens, H. (2010). Effects of inspiratory muscle training on respiratory function and repetitive sprint performance in wheelchair basketball players. British journal of sports medicine44(9), 665-668.

o   West, C. R., Taylor, B. J., Campbell, I. G., & Romer, L. M. (2014). Effects of inspiratory muscle training on exercise responses in Paralympic athletes with cervical spinal cord injury. Scandinavian journal of medicine & science in sports24(5), 764-772.

o   McDonald, T., & Stiller, K. (2019). Inspiratory muscle training is feasible and safe for patients with acute spinal cord injury. The journal of spinal cord medicine42(2), 220-227.

Answer: The duration of our study was short which could limit the efficacy of IMT. In those studies, the authors applied IMT during  6 weeks at 50% MIP(2 times a week: Goosey et al., 2010) or 6 weeks, at pressure threshold IMT (West et al., 2014); 50 sessions at 50% MIP (McDonald et al., 2019).  We acknowledge that the duration of our intervention may impact the effectiveness of inspiratory muscle training (IMT).

These studies suggest that a longer duration of IMT and/or the higher level of MIP for IMT may be necessary to achieve optimal results in respiratory muscle strength and function.

  1. Why authors used a subjective scale Numerical Rating Scale (NRS) for the inclusion criteria and not an objective scale e.g., MEP, MIP, FEV1, PEF, FVC, and FEF? Those would be more accurate and systematic.

  Answer: We agree but we give importance also the subjective evaluation of the SCI individuals together with objective assessments. Now, we put this points as a limitation of the study.

  1. It is recommended to have a sham group, implanting tSCS electrodes on the subjects without stimulation, to evaluate the effect of tSCS. Although, it would be hard to have blind subject, examiner and research staff should be blinded about sham and actual the treatment combination.

   Answer: We agree with the reviewer that is the ideal condition. Here we realized the study with control group because one of the principal reason was to reduce contact time with the SCI subjects during COVID pandemia and it was very difficult to realize sham stimulation, because SCI subjects experimented high intensity electrical stimulation for spinal motor potentials.

  1. As a suggestion, it would have been better to measure the effect of different frequencies and lower intensities of tSCS on the improvement of respiratory function (the excitation of phrenic interneurons/bulbospinal neurons). Lower intensity may also show beneficial effects and patient tolerance while reducing the probable side effects. Were there any side effects noted by the patients? Please add this information and discussion point about choosing the high intensity stimulation.

o   DiMarco, A. F., & Kowalski, K. E. (2013). Spinal pathways mediating phrenic activation during high-frequency spinal cord stimulation. Respiratory physiology & neurobiology, 186(1), 1-6.

o   DiMarco, A. F., Kowalski, K. E., Geertman, R. T., & Hromyak, D. R. (2009). Lower thoracic spinal cord stimulation to restore cough in patients with spinal cord injury: results of a National Institutes of Health–sponsored clinical trial. Part I: methodology and effectiveness of expiratory muscle activation. Archives of physical medicine and rehabilitation90(5), 717-725.

o    

  Answer: Thank you for highlighting the importance of studying the effect of different frequencies of tSCS and the intensity of stimulation. These factors play a crucial role in determining the therapeutic outcomes of tSCS interventions.

In our study, reported that applying tSCS at 90% motor threshold (MT) for the APB muscle was more effective  than 110% or 80% MT in promoting motor function, as well as spinal and cortical excitability in healthy subjects (Kumru H, Transcutaneous Electrical Neuromodulation of the Cervical Spinal Cord Depends Both on the Stimulation Intensity and the Degree of Voluntary Activity for Training. A Pilot Study J Clin Med. 2021). This finding suggests that the intensity of tSCS stimulation can influence its therapeutic effects on motor function and neural excitability .

Investigating the optimal frequency of tSCS is also essential. Different frequencies of stimulation may have differential effects on neuronal activity, synaptic plasticity, and functional outcomes.

We added next sentence in the conclusion: " Additionally, it is important to explore and understand the optimal parameters of tSCS, including intensity, frequency, and stimulation at different segments of the spinal cord. Tailoring the tSCS intervention to individual needs and optimizing the stimulation parameters may contribute to maximizing the therapeutic benefits and improving clinical outcomes for SCI individuals."

Overall, while our study contributes to the growing body of knowledge on the potential benefits of tSCS, further investigations are necessary to determine the most effective stimulation protocols and to better understand the underlying mechanisms of action. By continuing to explore and refine tSCS techniques, we can advance the field and ultimately improve outcomes for individuals with spinal cord injury.

We did not cite those articles because there are about invasive SCS and we studied the non-invasive SCS, but we added those references in the discusion.

  1. It is important to note that the only differences detected were in measurements that were self-reported (The subjects evaluated their sensation of breathlessness/dyspnea and hypophonia compared to pre-SCI using a numerical rating scale.) and no significant effects of the treatment was seen in any of the measured metrics. This should be discussed.  

  Answer: There was also significant improvement in the objective measurements such as maximal inspiratory and expiratory pressure (thorax inspiratory and expiratory muscle strength) and , pulmonar capacity ( Forced vital capacity). Please see it in the result sections and figures 2, 3 and 4 and in the table 2.

Specific Comments

  1. Abstract, the acronym “FVC” should be spelled out.

  Answer: Done as suggested.

  1. Line 137; the authors mentioned “young” patients. Was there a classification of young and old patients?

  Answer: We added range of age.(18-23years).

  1. There were several spots where “dyspnea” was misspelled. Please find them and edit them appropriately.

  Answer: Done as suggested

  1. Why authors used a low threshold NRS (≥ 3) for scoring the sensation of breathlessness/dyspnea for the inclusion criteria? (Subjective scale)

Answer: Breathlessness/dyspneais one of the important symptoms of respiratory problem following SCI   , which is underestimated. We added some information in the introduction.

We included breathlessness/dyspnea as a main inclusion complaints following injury and the mean of breathlessness was 5 point which is moderate in severity.  

We added this part as a limitation of the study.

  1. Stimulation location: authors described the position of the electrodes (cathode and anode) for the first time in the discussion section (page 11 line 302). It’s recommended to report this information in the Materials and Methods section.

Answer: it was in the methodology section- 2.7. Transcutaneous spinal cord stimulation  (page 5): " tSCS was carried out with a"BioStim-5" (Cosyma Inc., Moscow, Russia). Stimulation was delivered at C3-C4 and at Th9-Th10 using 2 cm diameter hydrogel adhesive electrodes as cathodes (Axelgaard, ValuTrodeâCloth) and two 5x12 cm2rectangular electrodes (Axelgaard, ValuTrodeâCloth) placed symmetrically over the iliac crests as anodes"

  1. Did the authors use  single electrical stimulation, or do you mean single pulse? (page 3 line 137)? I thought it was biphasic (according to line 308).

Answer: The EMG recording done in the muscles with single pulse electrical stimulation. The forma of pulse was biphasic in APB muscle and monphasic in diaphragm and abdominal muscles.

We rewrote this part to avoid confusion.

  1. What was the pulse width used for each 10K carrier for tSCS?

Answer: Yes. The mono or biphasic pulse width was 1ms and 10 kHz carrier is characteristic of a single electrical pulse. This information was specified morein the methodology in " 2.4. Neurophysiological assessment: and 2.7. Transcutaneous spinal cord stimulation".

  1. Based on the text it’s unclear whether the authors used biphasic or monophasic for tSCS? On page 3 line 137 they mentioned monophasic, while on page 4 line 185 they reported using biphasic for tSCS.

Answer: Now, we clarified it in the text: " Single 1 ms rectangular monophasic pulses were used to record spinal motor potential in the diaphragm and abdominal muscles, and biphasic pulses were used to record spinal motor potential in the APB muscle and then for tSCS at C3-4 and at Th9-10."

  1. If authors used monophasic for tSCS, what was the reason for that? And was it positive or negative monophasic?

Answer: We used biphasic for tSCS. Single-pulse monophasic stimulation was used only to record spinal motor potential in the diaphragm and abdominal muscles, which was not possible. For tSCS, we used biphasic stimulation (in the methodology in " 2.7. Transcutaneous spinal cord stimulation).  Because monophasic tSCS was more painful and the SCI subjects did not tolerate it very well especially at C3-4 segment. It was positive monophasic pulse. 

  1. In addition, did the authors use a single pulse (page 3 line 137) or 10K carrier (page 4 line 186) stimulation for tSCS?

Answer:We always used the same characteristics for transcutaneous spinal cord stimulation. tSCS waveform was biphasic, rectangular, 1 ms pulses at a frequency of 30 Hz, and filled with a carrier frequency of 10 kHz. The 10 kHz carrier is characteristic of a single electrical pulse and is typically much higher than the frequency of the actual stimulation pulses. It serves as a carrier wave that carries the lower-frequency stimulation pulses.

  1. Where authors placed/implanted recording electrodes?

Answer: The electrodes were placed in the parasternal line to record the spinal motor response of the diaphragm, and single-pulse electrical stimulation was applied at the C3-4 level. For recording the spinal motor response of the rectus abdominis, the electrodes were placed at the umbilical level, and single-pulse electrical stimulation was applied at the Th9-10 level.

Now, we added more information in the methodology: 2.4. Neurophysiological assessment.

  1. Why did the authors not evaluate tSCS alone? It could be more meaningful to compare tSCS and IMT individually. Please add to the discussion.

Answer: We reported that combining tSCS with muscle activity has demonstrated a greater ability to modulate the muscle force, the spinal cord and cortex excitability compared to using either intervention alone. This suggests that the combined approach may have synergistic effects in promoting neural modulation and plasticity (Kumru H, Flores A, et al. Cervical Electrical Neuromodulation Effectively Enhances Hand Motor Output in Healthy Subjects by Engaging a Use-Dependent Intervention . J. Clin. Med.  2021).

This part was added in the discussion.

  1. Fig 1. There is nothing in some of the boxes in this figure. Please review and edit!

Answer: We are sorry for these errors in the figures. Now we corrected them in the figures.

  1. Fig 2. This figure is very hard to read.  There are also misplaced texts in the figure that is on top of other texts making it impossible to read. Please edit. Figure legend for Fig 2 is also incomprehensible. The different bars are also not labelled. Please edit. P values should be in the text in the legend. It is hard to decipher what the p values are from if it is floating around in the figure. Y axis labels are unclear!

Answer: We are sorry for these errors in the figures. Now we corrected them in the figures.

  1. Fig 3. The legend states “A” and “B” but there is no “A” or “B” in the figure. What does “Max” refer to in the figure? Again, the p values are floating all around the figure and it is unclear what those numbers are associated with. Please edit. Y axis labels are unclear or not present!

Answer: We are sorry for these errors in the figures. Now we corrected them in the figures.

  1. Fig 4. The same problems as seen in Figures 2 & 3. Please edit the figure and figure legend thoroughly so that they can be legible.

Answer: We are sorry for these errors in the figures. Now we corrected them in the figures.

  1. Change “his” (page 11 line 321) to “this”.

Answer: Done as suggested.

  1. There were multiple places in the manuscript with extra spaces. Please review and edit.

Answer: Done as suggested

Reviewer 3 Report

In order to improve the manuscript, several changes must be included in the final version:

INTRODUCCIÓN

Authors should include information on the incidence and/or prevalence of Impaired respiratory function in subjects with spinal cord injury. Please add this relevant information in the final version of the manuscript.

Of all the methods described by the authors in the introduction to improve the respiratory function of subjects with spinal cord injury, which of them are currently the most widely used? Which is the most used in the Gutmann Institute? This information should be added in the final version of the manuscript.

The authors indicate "Based on previous findings of tSCS effects on motor function", to which previous studies do they refer in the previous sentence? Please describe these studies in more detail and include the corresponding bibliographical citations.

MATERIALS AND METHODS

Indicate the reference of the file approved by the ethics committee and the date of approval. Include this information in the final version of the manuscript.

It would be convenient to describe the parameters evaluated, since many non-expert readers do not have to know what the evaluation of the different spirometry parameters means from a clinical point of view. Please include a very brief description of each parameter in the final version of the manuscript.

RESULTS

The font of lines 242-247 is not the normative of the journal.

In figure 2 there is superimposed text that makes it difficult to read.

DISCUSSION

The discussion that the authors carry out in the manuscript is quite short. What is the novelty of the study carried out compared to previous similar studies? What is the clinical relevance of the results presented in the manuscript? Are there complications of the technique used? If so, what are these complications and what has been done to solve them? In how many world hospitals with patients suffering from spinal cord injury and respiratory disorders is the technique described in the manuscript used? The authors can also discuss electrical neuromodulation techniques applied to the respiratory system of patients with spinal cord injury and the potential plastic changes that occur in the electrically stimulated spinal cord that potentially explain the observed results. All these can be relevant points that the authors can include in the final version of the manuscript.

In case there is scientific information, the authors can also discuss these clinical results with similar results in animal models.

Author Response

We thank the reviewer for her/his constructive comments.

Authors should include information on the incidence and/or prevalence of Impaired respiratory function in subjects with spinal cord injury. Please add this relevant information in the final version of the manuscript.

Answer: We added more information on the incidence and/or prevalence of Impaired respiratory function in subjects with spinal cord injury in the introduction.

Of all the methods described by the authors in the introduction to improve the respiratory function of subjects with spinal cord injury, which of them are currently the most widely used? Which is the most used in the Gutmann Institute? This information should be added in the final version of the manuscript.

 Answer: We widely use inspiratory and/expiratory respiratory therapy and in more severe SCI patients, we implant the diaphragmatic pacemaker if indicated. We added this information in the the introduction.

The authors indicate "Based on previous findings of tSCS effects on motor function", to which previous studies do they refer in the previous sentence? Please describe these studies in more detail and include the corresponding bibliographical citations.

 Answer: We added more details about those studies and cited the references.

MATERIALS AND METHODS

Indicate the reference of the file approved by the ethics committee and the date of approval. Include this information in the final version of the manuscript.

Answer: It was at the end of the text "Institutional Review Board Statement:". Now, we also added it in the methodology.

It would be convenient to describe the parameters evaluated, since many non-expert readers do not have to know what the evaluation of the different spirometry parameters means from a clinical point of view. Please include a very brief description of each parameter in the final version of the manuscript.

Answer: The different spirometry parameters were explained more in the "Assessments of respiratory function"

RESULTS

The font of lines 242-247 is not the normative of the journal.

Answer: We changed it according to the normative of the journal.

In figure 2 there is superimposed text that makes it difficult to read.

 Answer: We are sorry for this error. We corrected all figures, which were modified according to journal format.

DISCUSSION

The discussion that the authors carry out in the manuscript is quite short. What is the novelty of the study carried out compared to previous similar studies?

Answer: We tried to expand the discussion a bit and give more information.

 What is the clinical relevance of the results presented in the manuscript?

Answer: The clinical implications of these findings are significant as they provide evidence for the effectiveness of tSCS combined with respiratory training as a potential therapeutic intervention for individuals with cervical SCI. Healthcare professionals involved in the management of individuals with SCI, including physicians, physiotherapists, and respiratory therapists, can consider incorporating this combined approach into their treatment plans to optimize respiratory function and improve outcomes in this population.

Are there complications of the technique used?

Answer: Seven SCI individuals complained of mild to moderate pain, but no one left the study. This information has been added to the last part of the result. 

If so, what are these complications and what has been done to solve them?

Answer: There were only complaints of pain. To avoid or reduce this, we started tSCS at low intensity, increasing it up to adequate intensity in 2-3 min.

In how many world hospitals with patients suffering from spinal cord injury and respiratory disorders is the technique described in the manuscript used?

Answer:I cannot provide specific information about the number of world hospitals using tSCS in respiratory disorders, but as we know there is a just one publication which is one case reported improvement in respiratory function in SCI individual with tSCS (Gad P, et al.Enabling respiratory control after severe chronic tetraplegia: an exploratory case study  J Neurophysiol. 2020). The field of tSCS is still relatively new, and its clinical use is not as widespread as other noninvasive stimulation techniques.

The authors can also discuss electrical neuromodulation techniques applied to the respiratory system of patients with spinal cord injury and the potential plastic changes that occur in the electrically stimulated spinal cord that potentially explain the observed results. All these can be relevant points that the authors can include in the final version of the manuscript.

 Answer: We added more discussion about electrical neuromodulation techniques applied to the respiratory system following SCI in the discussion

In case there is scientific information, the authors can also discuss these clinical results with similar results in animal models.

 Answer: We now discussed more the animal models and spinal cord stimulation.

Reviewer 4 Report

The authors provided a piece of evidence on the improvement of the respiratory function in patients with cervical spinal cord injury (cSCI) following tSCI (transcutaneous spinal cord stimulation) applied to C3-4 and Th9-10 neuromeres at a frequency of 30Hz for 30 minutes on 5 consecutive days.

However, some flaws should be depicted, making the paper necessary to be improved. 

1. Results are presented almost well, despite a short treatment time and a small number of participants. Eleven subjects received inspiratory muscle rehabilitation alone. Another eleven cSCI subjects received tSCS combined with rehabilitation; six of these subjects underwent rehabilitation alone first and then they were given the opportunity to receive tSCS and rehabilitation. The latter proposal in the study design is confusing, and seems to be a mistake; in fact, it makes the results unclear, and it is not clearly  known what provided the improvement in respiration.

2. In Keywords …transcutaneous electrical spinal cord stimulation; cervical spinal cord injury; respiratory function; transcutaneous spinal cord stimulation; respiratory rehabilitation… - the „ electrical stimulation” is unnecessary doubled.

3. The hypothesis and aim should be formulated more clearly in lines 72-80.

4. Figure 1 - what does mean the blank square on the top, similarly in the middle and the mysterious „enrollment” on the bottom? The legend of the figure resembles the draft.

5. The description of groups in lines 227-235 in M&M section is confusin, and should be rewritten.

6. Table 1 as well as Table 2 are  far away from the fashion of MDPI style.

7. Figure 2 is unclear.

8. Because of the short treatment time and the small number of participants, Conclusions should be presented more carefully.

9.  References are not in accordance with MDPI style and they are „economical” on the topic, and need upgrading.

Minor editing of English language required

Author Response

  1. The authors provided a piece of evidence on the improvement of the respiratory function in patients with cervical spinal cord injury (cSCI) following tSCI (transcutaneous spinal cord stimulation) applied to C3-4 and Th9-10 neuromeres at a frequency of 30Hz for 30 minutes on 5 consecutive days.

    However, some flaws should be depicted, making the paper necessary to be improved. 

    1. Results are presented almost well, despite a short treatment time and a small number of participants. Eleven subjects received inspiratory muscle rehabilitation alone. Another eleven cSCI subjects received tSCS combined with rehabilitation; six of these subjects underwent rehabilitation alone first and then they were given the opportunity to receive tSCS and rehabilitation. The latter proposal in the study design is confusing, and seems to be a mistake; in fact, it makes the results unclear, and it is not clearly  known what provided the improvement in respiration.

    Answer: In this version, we improved those points as suggested.

  2. In Keywords …transcutaneous electrical spinal cord stimulation; cervical spinal cord injury; respiratory function; transcutaneous spinal cord stimulation; respiratory rehabilitation… - the „ electrical stimulation” is unnecessary doubled.

Answer: Sorry for this error. Now the repeated words were eliminated as suggested.

  1. The hypothesis and aim should be formulated more clearly in lines 72-80.

Answer: We reformulated the hypothesis and aim as suggested.

  1. Figure 1 - what does mean the blank square on the top, similarly in the middle and the mysterious „enrollment” on the bottom? The legend of the figure resembles the draft.

Answer: Those errors were eliminated in the figures and we prepared new figures.

  1. The description of groups in lines 227-235 in M&M section is confusin, and should be rewritten.

Answer: We rewrote this section as suggested

  1. Table 1 as well as Table 2 are far away from the fashion of MDPI style.

Answer: We changed Table according to MDPI style.

  1. Figure 2 is unclear.

Answer: Now, Figure 2 was corrected in the last version of manuscript.

  1. Because of the short treatment time and the small number of participants, Conclusions should be presented more carefully.

 Answer: We modified the conclusion.

  1. References are not in accordance with MDPI style and they are „economical” on the topic, and need upgrading.

Answer: References were reviewed and corrected according to MDPI style.

Round 2

Reviewer 1 Report

The concern has not been addressed by the authors.

fine

Reviewer 3 Report

The authors have made an effort to respond to all of the reviewer's questions and suggestions. The quality of the manuscript has been significantly improved.

Reviewer 4 Report

The Authors answered the majority of my remarks well and the paper after the   Editorial Office support is suitable to be published in Biomedicines.

Minor correction during the work with the Academic Editor and the Type-setter